# Recent Advances of MXene-Based Electrochemical Immunosensors

**Meiqing Yang [1], Haozi Lu [2] and Song Liu [2,\*]**

1 Zoology Key Laboratory of Hunan Higher Education, College of Life and Environmental Science, Hunan University of Arts and Science, Changde 415000, China; meiqingyang2012@163.com

2 Institute of Chemical Biology and Nanomedicine (ICBN), State Key Laboratory of Chemo/Biosensing and Chemometrics, College of Chemistry and Chemical Engineering, Hunan University, Changsha 410082, China; luhaozi@hnu.edu.cn

\* Correspondence: liusong@hnu.edu.cn

**Abstract:** Electrochemical immunosensors are the largest class of affinity biosensing devices with strong practicability. In recent years, MXenes have become hotspot materials of electrochemical biosensors for their excellent properties, including large specific surface area, good electrical conductivity, high hydrophilicity and rich functional groups. In this review, we firstly introduce the composition and structure of MXenes, as well as their properties relevant to the construction of biosensors. Then, we summarize the recent advances of MXenes-based electrochemical immunosensors, focusing on the roles of MXenes in various electrochemical immunosensors. Finally, we analyze current problems of MXenes-based electrochemical immunosensors and propose an outlook for this research field.

**Keywords:** MXenes; electrochemical immunosensors; electrode-modified nanomaterials; labels

## 1. Introduction

Electrochemical immunosensors are the largest class of affinity biosensing devices for protein detection. Similar to the other biosensors, the electrochemical immunosensor consists of a biorecognition element, a transducer and a signal readout system. In a classical electrochemical immunosensor, the analyte (antigen) is specifically recognized through the capture antibody ($Ab_1$) coated on the electrode surface in the biorecognition element. Then, a recognition signal is generated on the electrode surface and converted into an electrical signal (current, voltage, resistance, etc.) through the transducer. Finally, a readable signal can be displayed on the readout system, so as to realize the purpose of analyte detecting.

Based on the types of electrical signal converted by the transducer, electrochemical immunosensors can be classified into amperometric, potentiometric, conductometric, impedimetric and capacitive sensors [1–4]. Based on the structural design, electrochemical immunosensors can be generally divided into non-labeled (label-free) and labeled (sandwich-type) immunosensors [5–7]. The non-labeled electrochemical immunosensor can detect the analyte through the change in the electrochemical impedance signal caused by antigen–antibody binding. The labeled electrochemical immunosensor usually requires a suitable electroactive label (enzyme, nanoparticle, etc.), which is conjugated to the detection antibody ($Ab_2$). The immuno-complex can be finally detected with the electroactive label via enzymatic or catalytic reaction, thus reflecting the level of analyte. Combing electrochemical analysis with antigen–antibody reaction, the electrochemical immunosensor has several distinct advantages, such as high specificity, excellent sensitivity, operational simplicity and low cost [8–11], and has received extensive attention in several fields such as the early diagnosis of disease, clinical analysis, food safety and environmental monitoring [12–14].

The performance of electrochemical biosensors largely depends on the sensing materials. The introduction of nanomaterials in the sensing system has promoted the development of electrochemical biosensors. To date, a variety of nanomaterials have been

synthesized and used to fabricate electrochemical biosensors. Among these materials, two-dimensional (2D) layered materials have attracted significant interest because of their large specific surface area, tunable electronic structure, excellent flexibility and mechanical properties [15–17]. The 2D layered materials, including graphene and transition metal dichalcogenides (TMDs) represented by $MoS_2$, have been widely applied in electrochemical biosensors. However, these 2D materials suffer from the problems of low electrical conductivity ($MoS_2$), high hydrophobicity ($MoS_2$, graphene) and difficulty in surface functionalization [18,19], which limit their further applications in electrochemical biosensors. With the development of nanomaterials, MXenes—as emerging graphene-like 2D layered functional nanomaterials—show the potential to overcome the problems above and become promising candidates for electrochemical biosensors.

To date, a certain amount of experimental works on MXenes-based electrochemical immunosensors have been reported. Furthermore, some of these works have been discussed in several review articles of MXene-based electrochemical sensors. For example, Kalambate et al. [18] reviewed MXenes-based electrochemical sensors for the detection of biomarkers, drugs, and environmental contaminants, which detailed an immunosensor using MXene as an electrode modifier for carcinoembryonic antigen detection. Mathew and Rout [20] presented the application progress of $Ti_3C_2T_x$ MXene in electrochemical biosensors and described the effect of MXene nanocomposites as electrode modifiers on the sensing performance of immunosensors. Wu et al. [21] summarized MXenes-based electrochemical sensors, which included four immunosensors. Yao et al. [22] summarized the recent progress of 2D MXenes nanomaterials in the construction of electrochemical enzyme-based biosensors, immunosensors and nucleic acid biosensors in 2021, which briefly discussed two immunosensors. However, the above-mentioned review articles only cover a small part of the work on MXenes-based electrochemical immunosensors, and do not provide a good generalization of the roles of MXenes in sensors, thus affecting the readers' comprehensive understanding of the field. Herein, we present the first comprehensive and systematic review on MXenes-based electrochemical immunosensors. First, we summarize the advances in the composition and structure of MXenes, as well as the properties of MXenes related to electrochemical biosensors. Then, we demonstrate different strategies of immunoassays and the corresponding roles that MXenes play in them. Finally, we also elucidate the current problems and propose suggestions for the future development of MXenes-based electrochemical immunosensors. We aim to inform readers of the recent advances of MXenes-based electrochemical immunosensors, and expect to provide helpful information for promoting the development of this field, so as to fabricate high-performance and practical electrochemical immunosensing devices for real-world human health, food safety and environmental monitoring.

## 2. Composition, Structure and Property of MXenes

MXenes are a family of transition metal carbides, nitrides or carbonitrides discovered by Gogotsi's group in 2011. MXenes are generally etched from the parent MAX phase with a hexagonal crystal structure and $P6_3/mmc$ symmetry [23]. The MAX phase can be represented by a general formula: $M_{n+1}AX_n$, where M is an early transition metal element (Ti, V, Zr, Nb, Mo, Hf, Sc, etc.), A is usually a group IIIA or IVA element (Al, Si, In, Pb, etc.), X is carbon or nitrogen element and *n* stands for 1, 2, 3, or 4 [24–26]. The M-X bond is stronger than the M-A bond [27,28]; thus, the A layer can be selectively removed from the MAX phase by an etchant to obtain MXene. The transition metal atoms (M) in the resulting MXene crystal are interlaced with carbon and/or nitrogen atoms (X), and the overall crystal maintains a hexagonal close-packed structure [29]. MXenes can be expressed with the general formula $M_{n+1}X_nT_x$, where $T_x$ is the surface functional group such as -OH, -O, -F, and -Cl. To date, more than 100 MXenes have been reported, of which more than 30 MXenes have been synthesized experimentally, and about 80 MXenes have been predicted by theoretical calculations [26]. Furthermore, all the above MXenes can be classified into four types in molecular structure: $M_2XT_x$, $M_3X_2T_x$, $M_4X_3T_x$ and $M_5C_4T_x$ [30].

The constituent elements and four typical molecular structures of MXenes are displayed in Figure 1.

**Figure 1.** Schematics of the constituent elements of MXenes in the periodic table (**top**) and four typical molecular structures of MXenes ($M_2XT_x$, $M_3X_2T_x$, $M_4X_3T_x$ and $M_5X_4T_x$) (**bottom**). Reprinted with permission from Ref. [24]. 2021, American Chemical Society.

Similar to other 2D materials, MXenes have a large specific surface area, which is beneficial for immobilizing biomolecules or adsorbing signal probes [31]. However, MXenes show several unique advantages. For example, MXenes have both excellent electrical conductivity [32] and tunable electronic structure [22]; thus, they can not only be used as sensing materials to accelerate electron transport, but also for fabricating special electronic devices. The abundant hydrophilic groups on the surface of MXenes endow them with a strong potential to combine with other active materials or biomolecules [33,34], which is beneficial for the construction and optimization of biosensors. MXenes are generally non-toxic and biocompatible with biological organisms, making them promising alternative biomaterials for versatile biomedical applications [23]. In addition, MXenes have good stretchability and can be used as conductive coatings for the fabrication of wearable electronics and biometric sensors [35]. Furthermore, MXenes can also be used as inks for printing, exhibiting the potential to realize miniaturization, integration and industrialization of MXenes-based devices [36]. In view of this, MXenes show promising prospects in the fabrication of high-performance biosensors. However, MXenes suffer from the inevitable issue of restacking and aggregation due to strong van der Waals forces between the layers [37,38], which affects their electrochemical performance. To address this issue, researchers tend to modify MXene nanosheets by introducing other active materials, which can act as not only spacers to prevent aggregation of MXene, but also promoters to enhance the electrochemical performance through synergistic effects. For example, researchers have introduced metal nanoparticles [39,40], carbon nanotubes [41,42], graphene [43,44], $MoS_2$ [45], conductive polymers [46] and ionic liquids [47] into MXenes to prepare interface materials with excellent electron transport properties, thereby further obtaining electrochemical sensors with good analytical performance.

## 3. Electrochemical Immunosensors Based on MXenes

With the increasing attention to MXenes, the applications of MXenes in electrochemical immunosensors have also been increasingly reported. According to these reports, MXenes can be used as electrode-modified nanomaterials (EMN), or as detection antibodies labels (Ab$_2$-label) to construct electrochemical immunosensors. Various electrical analysis meth-

ods are applied to detect analytes, including capacitance, differential pulse voltammetry (DPV), square wave voltammetry (SWV), amperometry, electrochemical impedance spectroscopy (EIS), and cyclic voltammetry (CV). These electrochemical immunosensors have been used to detect biomarkers of cancer, cardiovascular disease, sepsis and other diseases. The various electrochemical immunosensors are summarized in Table 1, including the MXene materials, etchant for MXenes preparation, roles, analytes, analytical methods and performance.

**Table 1.** Electrochemical immunosensors based on MXenes.

| MXene Material [Ref.] | Etchant for MXenes | Role | Analyte | Electrochemical Method | Limit of Detection | Dynamic Range |
|---|---|---|---|---|---|---|
| $f$-Ti$_3$C$_2$ [48] | HCl/LiF | EMN | CEA | CV | 0.018 pg mL$^{-1}$ | 0.0001–2000 ng mL$^{-1}$ |
| Ti$_3$C$_2$ [49] | HF | EMN | PSA | Capacitance | 0.031 pg mL$^{-1}$ | 0.1 ng mL$^{-1}$–50 ng mL$^{-1}$ |
| AuNPs/M-NTO-PEDOT [50] | HCl/LiF | EMN | PSA | DPV | 0.03 pg L$^{-1}$ | 0.0001–20 ng mL$^{-1}$ |
| Ti$_3$C$_2$-AuNP [51] | HF | EMN | PSA | DPV | 0.31 pg mL$^{-1}$ | 1 pg mL$^{-1}$–50 ng mL$^{-1}$ |
| AuNP-Ti$_3$C$_2$T$_x$ [52] | HF | EMN | CYFRA21-1 | SWV | 0.1 pg·mL$^{-1}$ | 0.5–10,000 pg·mL$^{-1}$ |
| MXNSs-AFBPB [53] | HCl/LiF | EMN | Apo-A1 NMP 22 | DPV | 0.3 pg mL$^{-1}$ 0.7 pg mL$^{-1}$ | 0.1 pg mL$^{-1}$–50 ng mL$^{-1}$ |
| $f$-Ti$_3$C$_2$ [54] | HCl/LiF | EMN | cTnI | DPV | 0.58 ng mL$^{-1}$ | 5–100 ng mL$^{-1}$ |
| AuNPs/Ti$_3$C$_2$T$_x$@PAMAM [9] | HF | EMN | cTnT | DPV | 0.069 ng mL$^{-1}$ | 0.1–1000 ng mL$^{-1}$ |
| Ti$_3$C$_2$T$_x$ [55] | HCl/LiF | EMN | Cortisol | EIS | 3.88 pM | 0.01–100 nM |
| L-cys/AuNP/Ti$_3$C$_2$T$_x$ [56] | HF | EMN | Cortisol | Amperometry | 0.54 ng mL$^{-1}$ | 5–180 ng mL$^{-1}$ |
| AuNPs/d-S-Ti$_3$C$_2$T$_x$ [57] | HCl/LiF | EMN | PCT | DPV | 2.0 fg mL$^{-1}$ | 0.01–1.0 pg mL$^{-1}$ |
| AuNP/Ti$_3$C$_2$T$_x$ [58] | HF | EMN | ALP | DPV | 4.0 U L$^{-1}$ | 10.0–1500 U L$^{-1}$ |
| Ti$_3$C$_2$T$_x$@AuNPs [59] | HCl/LiF | Ab$_2$-label | PSA | DPV | 3.0 fg mL$^{-1}$ | 0.01–1.0 pg mL$^{-1}$ |
| Ti$_3$C$_2$@CuAu-LDH [60] | HCl/LiF | Ab$_2$-label | CEA | SWV Amperometry | SWV: 33.6 fg mL$^{-1}$ i–t: 45.4 fg mL$^{-1}$ | 0.0001–80 ng mL$^{-1}$ |
| CuPtRh CNBs/NH$_2$-Ti$_3$C$_2$ [61] | HCl/LiF | Ab$_2$-label | cTnI | Amperometry | 8.3 fg mL$^{-1}$ | 25 fg mL$^{-1}$–100 ng mL$^{-1}$ |
| Cd$_{0.5}$Zn$_{0.5}$S/d-Ti$_3$C$_2$Tx [62] | HCl/LiF | Ab$_2$-label | h-FABP | DPV | 3.30 fg mL$^{-1}$ | 0.01–1.00 pg mL$^{-1}$ |
| PdPtBP MNPs/Ti$_3$C$_2$T$_X$ [63] | / | Ab$_2$-label | KIM-1 | DPV | 86 pg mL$^{-1}$ | 0.5 ng mL$^{-1}$–100 ng mL$^{-1}$ |

Abbreviation: $f$-Ti$_3$C$_2$—(3-Aminopropyl)triethoxysilane (APTES) functionalized Ti$_3$C$_2$-MXene; AuNP—gold nanoparticle; M-NTO—Ti$_3$C$_2$ MXene-derived sodium titanate nanoribbons; MXNSs—MXene-Ti$_3$C$_2$T$_x$ nanosheets; AFBPB—4-amino-1-(4-formyl-benzyl)pyridinium bromide; PAMAM—poly(amidoamine); L-cys—L-cysteine; d-S-Ti$_3$C$_2$T$_X$—sulfur-doped Ti$_3$C$_2$T$_X$; LDH—layered double hydroxide; CNBs—cubic nanoboxes; d-Ti$_3$C$_2$T$_x$—delaminated Ti$_3$C$_2$T$_x$ MXene; MNPs—mesoporous nanoparticles; CEA—carcinoembryonic antigen; PSA—prostate-specific antigen; CYFRA21-1—cytokeratin fragment antigen 21-1; Apo-A1—apolipoprotein-A1; NMP 22—nuclear matrix protein 22; cTnI—cardiac troponin I; cTnT—cardiac troponin T; PCT—procalcitonin; ALP—alkaline phosphatase; h-FABP—heart-type fatty acid-binding protein; KIM-1—kidney injury molecule-1.

### 3.1. MXenes as Electrode-Modified Nanomaterials in Electrochemical Immunosensors

The basis for fabricating a high-performance electrochemical biosensor is to construct a well-defined functional interface. Thus, EMN between the transducer and the target recognition layer is essential for efficient electron transport and immobilization of biomolecules [9,64]. Therefore, it is of great significance to explore novel EMN with excellent properties. As emerging 2D materials, MXenes have become ideal EMN for their excellent properties, such as high electrical conductivity, large specific surface area, and good biocompatibility. So far, electrochemical immunosensors fabricated with MXenes and their composite have been applied for the detection of various cancer biomarkers (CEA, PSA, CYFRA21-1, Apo-A1 and NMP 22), cardiovascular disease biomarkers (cTnI and cTnT), sweat biomarkers (cortisol), inflammation biomarkers (PCT) and so on.

Salama's group [48] first reported a label-free MXenes-based immunosensor for the detection of CEA (Figure 2), an important broad-spectrum tumor marker in clinical applications. Ti$_3$C$_2$-MXene nanosheets were treated with APTES to form functionalized Ti$_3$C$_2$-MXene ($f$-Ti$_3$C$_2$-MXene) for covalent immobilization of the capture antibodies (anti-CEA). Based on the effects of different redox probes, hexaammineruthenium ([Ru(NH$_3$)$_6$]$^{3+}$) was selected as the preferable redox probe. The as-prepared $f$-Ti$_3$C$_2$-MXene immunosensor exhibited a linear detection range of 0.0001–2000 ng mL$^{-1}$ with a limit of detection (LOD) of 0.018 pg mL$^{-1}$. This work opens up a new opportunity for the development of MXene-based biosensors.

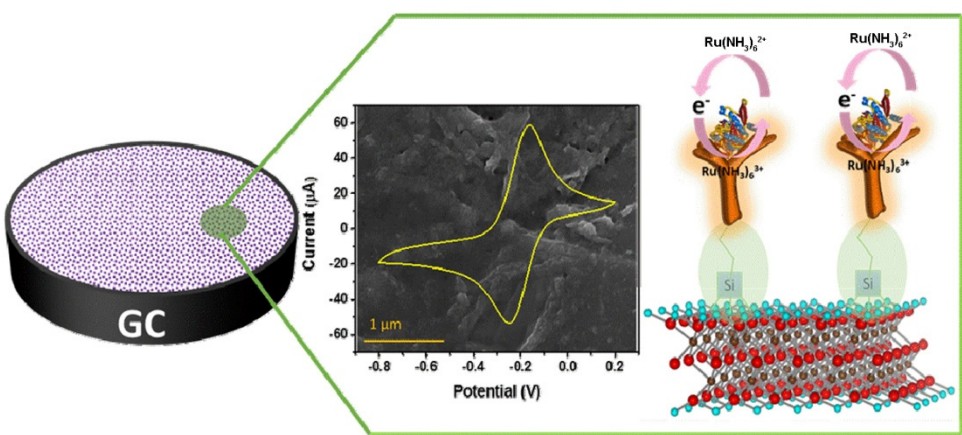

**Figure 2.** Schematic diagram of the mechanism for electrochemical CEA detection. Adapted with permission from Ref. [48]. 2018, Elsevier.

PSA detection is critical for clinical diagnosis of prostate cancer. Chen et al. [49] reported a horseradish peroxidase (HRP)-labeled immunosensor for the detection of PSA. The interdigitated micro-comb electrode was modified with $Ti_3C_2$ MXenes, which could immobilize anti-PSA capture antibody via π-stacking reaction. The detection antibody was prepared with HRP-labeled AuNPs as nanocarrier, which was further enhanced with tyramine-HRP repeats. After immuno-interaction, in the presence of $H_2O_2$, the tyramine-HRP oxidized 4-chloro-1-naphthol molecules into insoluble benzo-4-chlorohexadienone layer, leading to the local changes in capacitance. Combing $Ti_3C_2$ MXenes electrode-modified nanomaterials with tyramine signal amplification strategy, the capacitive immunosensors achieved sensitive detection of target PSA, and exhibited good reproducibility, higher specificity and accuracy for human serum analysis in comparison with a commercial PSA enzyme-linked immunosorbent assays (ELISA) kit. To overcome the restacking and agglomeration of MXenes, Xu and co-worker [50] synthesized 3D M-NTO by oxidizing and alkalizing $Ti_3C_2$ MXene nanosheets. After anchoring PEDOT and further electrodepositing AuNPs on the surface, the AuNPs/M-NTO-PEDOT composite was obtained and applied for EMN. Then, a label-free immunosensor was fabricated based on this composite for the quantitative determination of PSA. Since the large specific surface area of AuNPs/M-NTO-PEDOT composite can load a large number of capture antibodies, and its excellent electrical conductivity can promote charge transfer, the immunosensor exhibited excellent sensitivity, with a low LOD of 0.03 pg $L^{-1}$. Recently, a sandwich-type immunosensor for quantitative PSA screening was fabricated by integrating MXene ($Ti_3C_2$)−AuNPs as EMN and dopamine-loaded liposomes for signal amplification. The immunosensor also showed good analytical performance [51].

The efficient trace detection of CYFRA21-1 is critical for early diagnosis and prognosis of non-small cell lung cancer (NSCLC). Cheng et al. developed a sandwiched immunosensor for ultrasensitive detection of CYFRA21-1 [52]. Combing AuNPs decorated $Ti_3C_2T_x$-MXene (AuNP-$Ti_3C_2T_x$) as EMN to load numerous capture antibodies and accelerate the electron transfer rate, and toluidine blue (TB) modified AuNPs-doped covalent organic framework (COF) polymer as a detection antibody tag for amplifying the signal, the immunosensor showed excellent sensitivity for CYFRA21-1 detection and can be applied to real serum samples analysis.

Multiplex detection of tumor biomarkers is more reliable than single-antigen detection. Park group [53] reported a label-free immunosensor for multiplex detection of bladder cancer markers (Figure 3). Two-dimensional MXNSs were first controllably deposited onto a gold dual interdigitated microelectrode (DIDμE) via electroplating to enhance the active interface. A task-specific ionic liquid AFBPB, with excellent ionic conductivity, was subsequently modified onto MXNS surfaces via π-π and electrostatic adsorption to further covalently bind antibodies. Attributed to the synergistic electrochemical effect of MXNSs

and FBPB, the redox current of the MXNSs-AFBPB modified DIDμE was seven times higher than that of the bare electrode. Using Apo-A1 and NMP 22 as model analytes, the resulting MXNSs-AFBPB modified DIDμE immunosensor exhibited a wide linear response range (0.1 pg mL$^{-1}$ to 50 ng mL$^{-1}$) with LOD values as low as 0.3 and 0.7 pg mL$^{-1}$, respectively, and acceptable practicability in the detection of human urine samples.

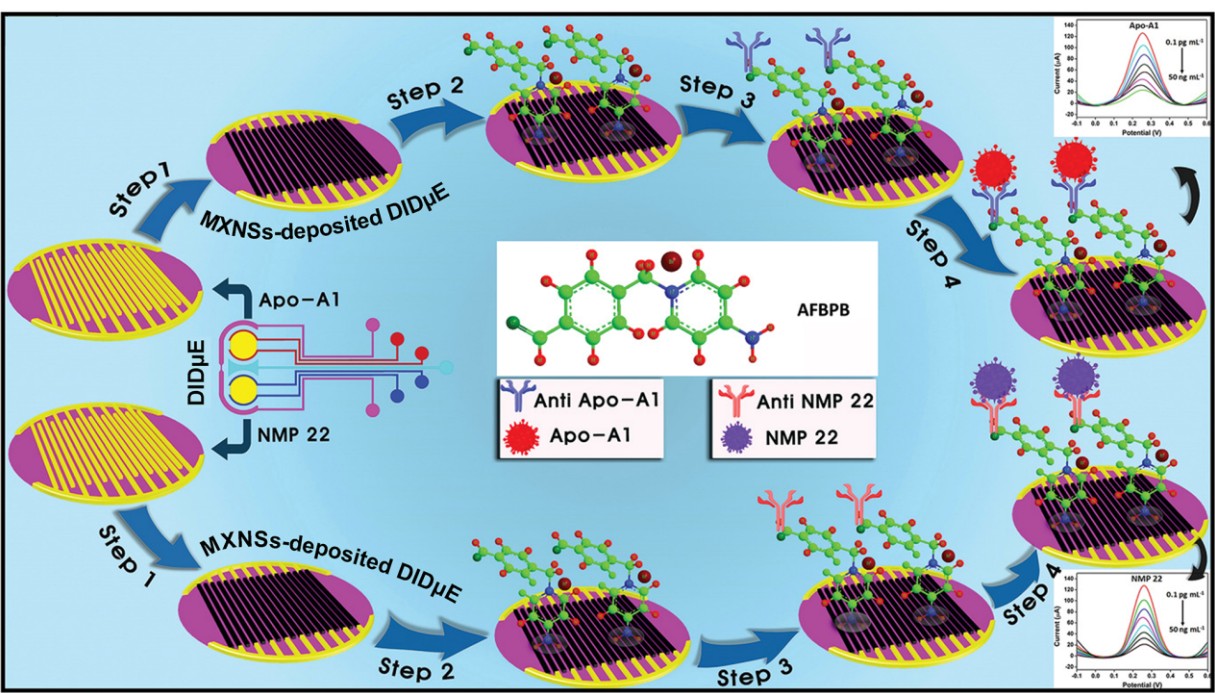

**Figure 3.** Schematic illustration of the fabricating procedure of the immunosensor for Apo–A1 and NMP 22 detection. Adapted with permission from Ref. [53]. 2020, Wiley–VCH GmbH.

MXene-based immunosensors for detecting biomarkers of cardiovascular disease have been reported. By modifying $f$-Ti$_3$C$_2$-MXene on the screen-printed carbon electrodes (SPCEs), Wang and co-worker [54] developed a label-free immunosensor for the detection of cTnI, one of the most specific biomarkers for acute myocardial infarction (AMI). Since the excellent conductivity and large surface area of $f$-Ti$_3$C$_2$-MXene facilitated both the electron transfer and antibody immobilization, the immunosensor achieved rapid and sensitive detection of cTnI, exhibiting great potential in clinical monitoring of AMI. Recently, another label-free immunosensor for the analysis of human cTnT was reported [9]. To overcome the problems of restacking and anodic oxidation of MXenes, 2D Ti$_3$C$_2$Tx nanosheets were covalently functionalized with first-generation Poly(amidoamine) (PAMAM) dendrimers to form MXene@PAMAM composite. The synergistic effect of MXene and PAMAM endowed the composite with good electrical conductivity, large specific surface areas and impressive electrochemical stability. Since massive amino terminals of PAMAM provided abundant active sites, AuNPs were subsequently self-assembled to construct AuNPs/MXene@PAMAM 3D heterogeneous nanoarchitecture. Benefiting from the structural and functional advantages of the AuNPs/MXene@PAMAM, the resulting immunosensor showed desirable stability, high specificity and good sensitivity for cTnT detection, with a LOD of 0.069 ng mL$^{-1}$.

MXene-based immunosensors for detecting sweat biomarkers have also been reported. A wearable microfluidic-integrated impedimetric immunosensor was developed for cortisol detection in human sweat, based on a Ti$_3$C$_2$T$_x$ MXene-laser-burned graphene (LBG) electrode [55]. The strategy of depositing highly conductive Ti$_3$C$_2$T$_x$ MXenes onto LBG electrode addressed the problem of inter-flake disconnection caused by burning and transferring, thereby improving the electrical properties of the LBG electrode. Integrated with a

microfluidic system, the prepared patch immunosensor showed good selectivity, as well as a low LOD of 3.88 pM. Based on L-cys/AuNPs/MXene modified conductive thread electrode, a label-free immunosensor was fabricated for non-invasive detection of sweat cortisol (Figure 4) to identify adrenal gland disorders [56]. Since L-cys-/AuNPs/MXene not only enhanced the electrochemical conductivity of the conductive thread electrode, but also increased the surface area for antibody immobilization, the immunosensor achieved high sensitivity, good reproducibility and long-term storage stability. Moreover, the immunosensor was successfully applied for cortisol detection in artificial sweat, with satisfactory performance.

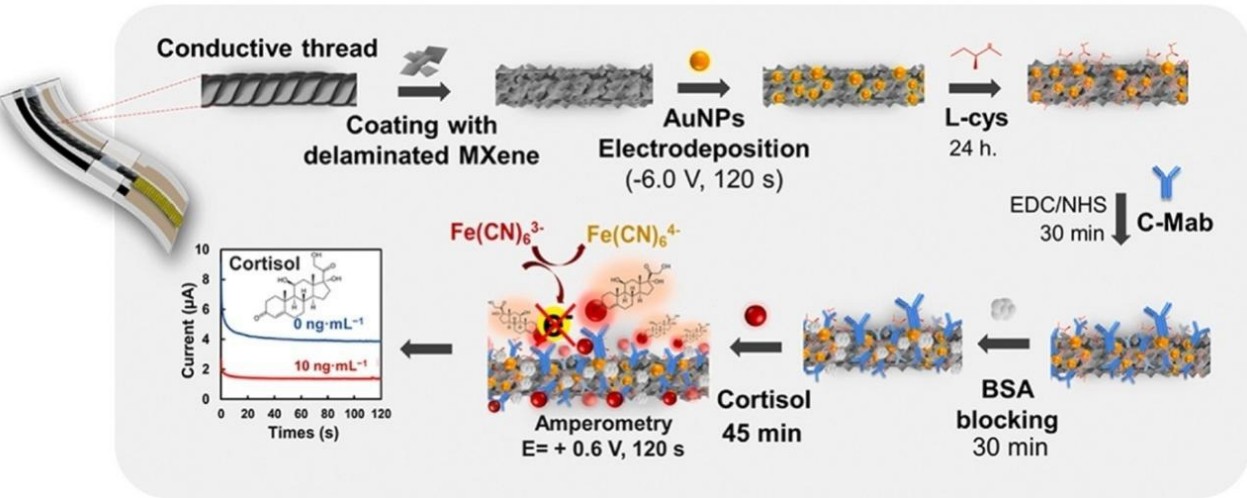

**Figure 4.** Schematic illustration of the fabrication steps of the thread-based electrochemical immunosensor for cortisol detection. Reprinted with permission from Ref. [56]. 2022, Elsevier.

Medetalibeyoglu et al. [57] reported a sandwich-type immunosensor for the detection of PCT, a key biomarker of septicemia. In their report, glassy carbon electrode (GCE) was modified with $AuNPs/d$-S-$Ti_3C_2T_X$ (gold nanoparticles functionalized sulfur-doped MXene) to increase surface conductivity and the amount of capture antibody. The c-g-$C_3N_4$ (carboxylated graphitic carbon nitride) was used to label detection antibody via a strong interaction between -COOH of c-g-$C_3N_4$ and $-NH_2$ of monoclonal detection antibody, thus achieving stable signal amplification. The resulting immunosensor realized an ultrasensitive detection of PCT, with a LOD of 2.0 fg mL$^{-1}$.

### 3.2. MXenes as Detection Antibody Label in Electrochemical Immunosensors

Early diagnosis of disease is an important goal of electrochemical immunosensors. The level of biomarkers in body fluids is often very low in the early stage of disease [7]; thus, a label-free immunosensor that increases the electrochemical response merely by modifying electrodes would be not sufficient for meeting detection needs. To achieve high-sensitivity detection of analytes, it is necessary to construct a sandwich-type immunosensor by labeling detection antibody with various signaling molecules to further increase the electrochemical response. MXenes have intrinsic good electrocatalytic activity and can be directly used as signaling molecules to label detection antibodies [59,65]. In addition, attributed to the large specific surface area and powerful ability to bind other signaling molecules, MXenes have been also used as excellent carriers for other signaling molecules.

Medetalibeyoglu et al. [59] designed a sandwich-type immunosensor for PSA detection (Figure 5). In their work, $Ti_3C_2T_X$ MXene@AuNPs, with good electrocatalytic activity for $H_2O_2$ redox reaction, was used as a signal molecule to label detection antibody for signal amplification. With AuNPs-ATPGO (gold nanoparticles/p-aminothiophenol functionalized graphene oxide) composite modified GCE as a sensing platform, the immunosensor obtained a LOD of 3.0 fg mL$^{-1}$. Zhang and co-worker [60] fabricated a sandwiched im-

munosensor for the quantitative detection of CEA. In their work, $Ti_3C_2$ MXenes were introduced into CuAu-LDH to improve the conductivity and facilitate the electron transfer between $Cu^{2+}$ and $Cu^+$. After that, the $Ti_3C_2$ MXenes-anchored CuAu-LDH 2D materials were used to label detection antibody. SWV and amperometry (i–t) dual-mode analysis were employed to reduce systematic errors and improve detection accuracy. The immunosensor finally achieved a low LOD (SWV: 33.6 fg $mL^{-1}$ and i–t: 45.4 fg $mL^{-1}$).

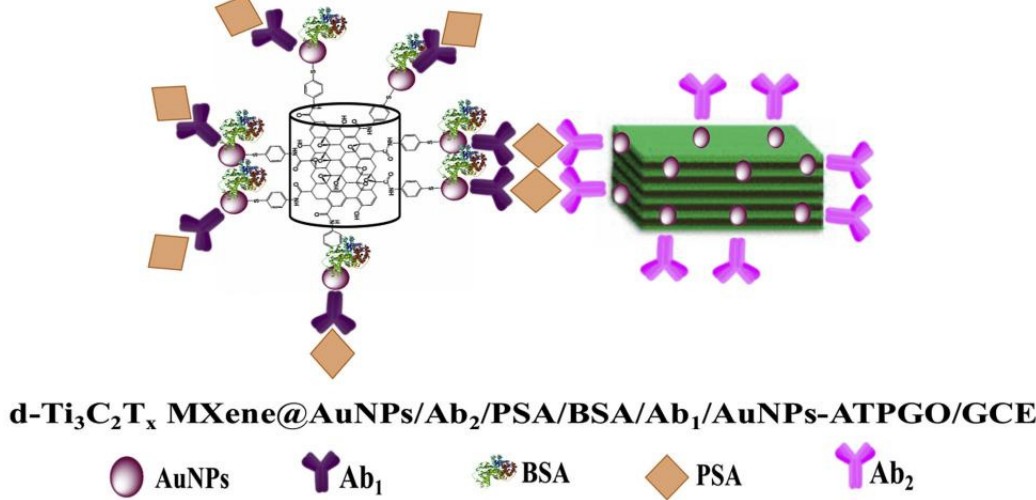

**d-$Ti_3C_2T_x$ MXene@AuNPs/$Ab_2$/PSA/BSA/$Ab_1$/AuNPs-ATPGO/GCE**

⬤ AuNPs     Y $Ab_1$    🌸 BSA    ◆ PSA    Y $Ab_2$

**Figure 5.** Schematic illustration of sandwich-type immunosensor for quantitative detection of PSA. Reprinted with permission from Ref. [59]. 2020, Elsevier.

For the early diagnosis of cardiovascular disease, a sandwich-type immunosensor was designed for cTnI detection [61]. CuPtRh CNBs/$NH_2$–$Ti_3C_2$, composed of trimetallic hollow CuPtRh cubic nanoboxes and few-layer ultrathin ammoniated MXene, was applied for detection antibody labels (Figure 6a). The embedded CuPtRh CNBs acted as not only a spacer to prevent the $NH_2$–$Ti_3C_2$ layer from restacking, but also a connector to immobilize more $Ab_2$ through stable Pt–N and Rh–N bonds. In addition, the CuPtRh CNBs provided more active sites for catalytic $H_2O_2$ reduction, thereby effectively amplifying the current signal of the immunosensor. The designed immunosensor exhibited acceptable reproducibility, stability and specificity for cTnI detection, with a LOD of 8.3 fg $mL^{-1}$. Karaman et al. [62] developed a sandwich-type immunosensor for the detection of h-FABP, another important biomarker in the diagnosis of AMI. To prepare the immunosensor, $Cd_{0.5}Zn_{0.5}S$/$Ti_3C_2T_x$ MXene nanocomposite was used to label detection antibody to amplify the signal. The $Cd_{0.5}Zn_{0.5}S$/$Ti_3C_2T_x$ MXene nanocomposite with large specific surface area not only facilitated the $Ab_2$ immobilization, but also promoted electron transfer, resulting from the remarkable synergistic effect between $Cd_{0.5}Zn_{0.5}S$ and $Ti_3C_2T_x$ MXene. Using the hc-g-C3N4@CDs (high-crystalline graphitic carbon nitride@carbon dots) core–shell nanostructure as sensing matrix, the prepared immunosensor indicated an ultra-short detection time and a high sensitivity, with a LOD of 3.30 fg $mL^{-1}$.

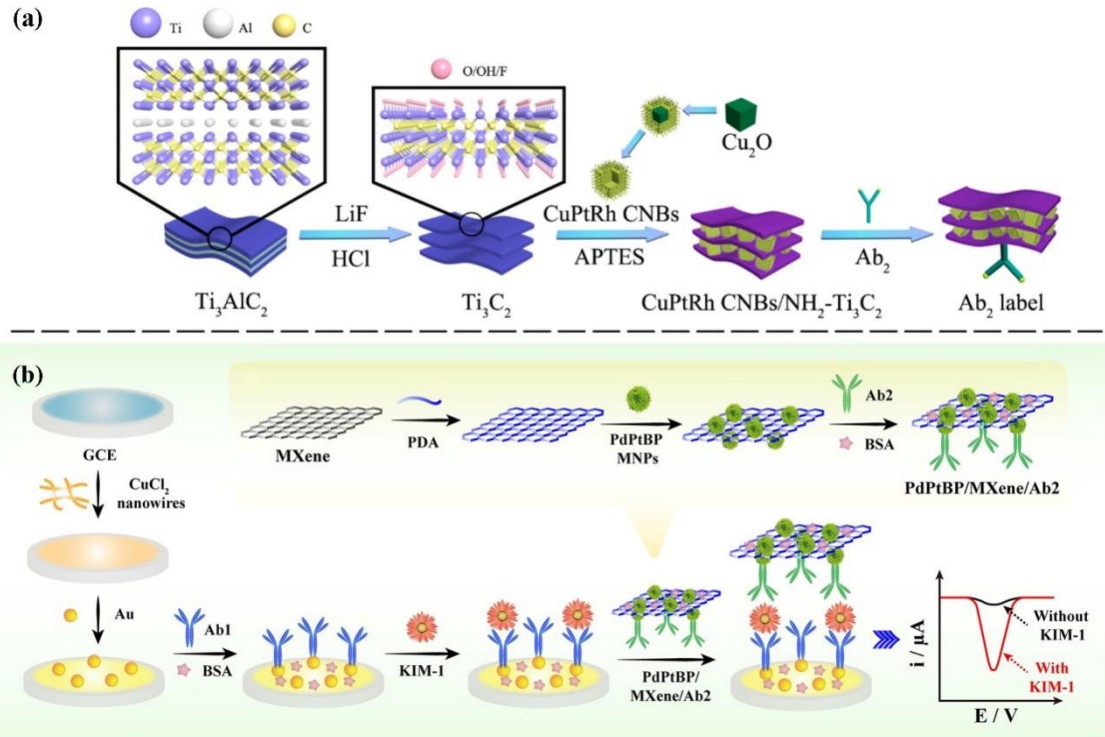

**Figure 6.** (**a**) The preparation steps of CuPtRh CNBs/NH$_2$–Ti$_3$C$_2$ based Ab$_2$ label for cTnI detection. Reprinted with permission from Ref. [61]. 2020, American Chemical Society. (**b**) The fabrication steps of sandwich-type electrochemical immunosensor for detection of KIM-1. Reprinted with permission from Ref. [63]. 2021, Elsevier.

Liu et al. [63] reported a sandwich-type immunosensor for ultrasensitive detection of KIM-1, a powerful biomarker of acute kidney injury (AKI) that can non-invasively reflect the kidney injury and recovery process. Specifically, PdPtBP MNPs/MXene nanocomposites were successfully synthesized using polydopamine (PDA) functionalized MXene with large surface area and the quaternary PdPtBP MNPs nanozymes (Figure 6b). Since the PdPtBP MNPs nanozymes had excellent peroxidase-like catalytic activity and could covalently bind to the antibodies, the PdPtBP MNPs/MXene nanocomposites were used as Ab$_2$ signal amplification labels. Meanwhile, the combination of highly conductive CuCl$_2$ nanowires and AuNPs was used as the sensing matrix to conjugate the Ab$_1$ onto the GCE surface. Benefiting from the high catalytic activity of PdPtBP MNPs nanozymes and the excellent conductivity of CuCl$_2$ NWs, the prepared immunosensor exhibited satisfactory performance for ultrasensitive detection of KIM-1 and can be successfully applied for the specific detection of KIM-1 in urine.

## 4. Conclusions and Outlook

In this review, recent advances in MXenes-based electrochemical immunosensors have been summarized, with a focus on the roles of MXenes as electrode-modified materials and detection antibody labels. Attributed to the excellent properties of MXenes, their application in electrochemical immunosensors have shown encouraging prospects. However, there is still much room for improvement in future research of MXenes-based electrochemical immunosensors to realize the practical applications. (1) Most of the MXenes for preparation of electrochemical immunosensors are synthesized by conventional HF etching. The residual -F group is not only harmful to the environment, but also affects the performance of the sensor. It is urgent to adopt environmentally friendly and efficient synthesis techniques for MXenes, such as alkali etching, electrochemical etching, molten salt etching, etc. [25]. (2) The MXenes used in the preparation of electrochemical immunosensors are

all $Ti_3C_2T_x$ with single performance. In the future, various transition metal carbides such as $Nb_2C$ [66], $Mo_4VAlC_4$ [67], and even transition metal nitrides with higher conductivity, such as $Ti_4N_3$ [68], can also be applied in electrochemical immunosensors for exploring multiple or synergistic performance. (3) Due to the strong interlayer van der Waals forces, the synthesis of uniform and well-dispersed MXenes remains a great challenge. In the future, interfacial electrochemical self-assembly [69] or other innovative approaches should be utilized to synthesize multi-dimensional MXenes to prevent their restacking and aggregation. (4) Although the multiplex detection electrochemical immunosensor has been fabricated [53], the signals from different analytes cannot be simultaneously displayed on the readout system, and a multiplex detection device has not been developed until now. In the future, using various redox probes [70], it is feasible to design multiple detection platforms that can simultaneously display the signals of different analytes on the readout system. Such platforms can achieve a faster response and further develop the corresponding point-of-care testing (POCT) devices in combination with circuit-board design and smart phones [71,72]. (5) The current MXene-based electrochemical immunosensors are mainly used for disease diagnosis. In the future, the application scope of MXene-based electrochemical immunosensors can be extended to the environment, agriculture and other fields. In conclusion, the MXenes-based electrochemical immunosensors have promising prospects. The research of MXenes is in the ascendant. It is expected that with the in-depth research on MXenes, high-performance MXene-based electrochemical immunosensors will be developed, thus truly contributing to the production and life of human beings.

**Author Contributions:** Conceptualization, M.Y. and S.L.; data curation, M.Y. and H.L.; supervision, S.L.; writing—original draft, M.Y.; writing—review and editing, M.Y. and S.L. All authors have read and agreed to the published version of the manuscript.

**Funding:** This work was supported by the National Natural Science Foundation of China (No. 22175060, 21975067), the Natural Science Foundation of Hunan Province of China (2021JJ10014, 2021JJ30092) and the Doctoral Research Start-up Fund of Hunan University of Arts and Sciences (21BSQD43).

**Institutional Review Board Statement:** Not applicable.

**Informed Consent Statement:** Not applicable.

**Data Availability Statement:** Not applicable.

**Conflicts of Interest:** The authors declare no conflict of interest.

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
