# Peer review of "Recent Advances of MXene-Based Electrochemical Immunosensors"

_applsci, doi:10.3390/app12115630_

Round 1

Reviewer 1 Report

Dear Editor,

The manuscript entitled “Recent Advances of MXene-based Electrochemical Immunosensors” by Meiqing Yang et al. summarize the recent advances of MXenes-based electrochemical immunosensors. The authors report the advances in composition and structure of MXenes, and the properties of MXenes related to electrochemical biosensors. They also presented MXenes-based electrochemical immunosensors, focusing on analyzing the role of MXenes, and they summarized current problems and future outlook of MXenes-based electrochemical immunosensors.

In my opinion, the manuscripts’ objective and perspective are quite interesting, the manuscript is very well structured and well-written. The inclusion of figures is very helpful to the reader and even though several reviews have been published on the subject, the objective of the present manuscript is distinct and fills a gap in Xenes-based electrochemical immunosensors. In my opinion, the authors should add some possible solutions in the current problems on MXenes-based electrochemical immunosensors. Therefore, I think that the manuscript should be accepted for publication after minor revisions.

Author Response

In my opinion, the authors should add some possible solutions in the current problems on MXenes-based electrochemical immunosensors. Therefore, I think that the manuscript should be accepted for publication after minor revisions.

Response: Thanks for your suggestion. We have added some possible solutions to current problems on MXenes-based electrochemical immunosensors in the Conclusion and outlook section, which are shown in the revised manuscript.

Reviewer 2 Report

Re: Recent Advances of MXene-based Electrochemical Immunosensors

Manuscript ID: applsci-1732522

This paper summarize the recent advances of MXenes-based electrochemical immunosensors, focusing on the roles of MXenes invarious electrochemical immunosensors. This manuscript may be suitable for publication in Applied Sciences after major revision. Some suggested notes mentioned below may help authors to develop their work.

  • The language should be revised. For example:
  • line 16: invariousshould be in various
  • line 18: analyzecurrent should be analyze current
  • line 25: immunosensorconsists should be immunosensor consists
  • line 28: Then should be element. Then
  • line 30:Finally should be transducer. Finally
  • line 37: immunosensorcan should be immunosensor can
  • line 38: signalcaused should be signal caused
  • line 39: immunosensorusually should be immunosensor usually
  • line 40: “(Ab2).The “ should be “(Ab2). The “
  • line 41: finallydetected should be finally detected
  • line 42: “Combingelectrochemical” should be Combing electrochemical
  • line 43: “immunosensorhasseveral “ should be “immunosensor has several”
  • line 46: “safety, environmental” should be “safety, and environmental”
  • line 48: “systemhas” should be “system has”
  • line 69: “worldhuman” should be “world human”
  • line 85: “bytheoretical” should be “by theoretical”
  • line 93: “alsoused” should be “also used”
  • line 98: “thempromising” should be “them promising”
  • line 103: “fabricationof” should be “fabrication of”
  • line 111: “[43], ionic” should be “[43] and ionic”
  • line 123: “methodsare” should be “methods are”
  • line 142: “interface.Thus,” should be “interface. Thus,”
  • line 143: “are” should be “is”
  • line 147: “immunosensorsfabricated” should be “immunosensors fabricated”
  • line 148: “appliedfor” should be “applied for”
  • line 169: “at” should be “in”
  • line 179: “were” should be “was”
  • line 182: “amount” should be “number”
  • line 197: “multipex” should be “multiplex”
  • line 202: “tothe” should be “to the”
  • line 215: “biomarker” should be “biomarkers”
  • line 232: “cortisoldetection” should be “cortisol detection”
  • line 237: “alow” should be “a low”
  • line 237: “showedgood” should be “showed good ”
  • line 244: “thecortisol detectionin” should be “the cortisol detection in”

and so on…..

  • The review needs careful organization.

Author Response

(1) The language should be revised. For example:

line 16: invariousshould be in various

line 18: analyzecurrent should be analyze current

line 25: immunosensorconsists should be immunosensor consists

line 28: Then should be element. Then

line 30:Finally should be transducer. Finally

line 37: immunosensorcan should be immunosensor can

line 38: signalcaused should be signal caused

line 39: immunosensorusually should be immunosensor usually

line 40: “(Ab2).The “ should be “(Ab2). The “

line 41: finallydetected should be finally detected

line 42: “Combingelectrochemical” should be Combing electrochemical

line 43: “immunosensorhasseveral “ should be “immunosensor has several”

line 46: “safety, environmental” should be “safety, and environmental”

line 48: “systemhas” should be “system has”

line 69: “worldhuman” should be “world human”

line 85: “bytheoretical” should be “by theoretical”

line 93: “alsoused” should be “also used”

line 98: “thempromising” should be “them promising”

line 103: “fabricationof” should be “fabrication of”

line 111: “[43], ionic” should be “[43] and ionic”

line 123: “methodsare” should be “methods are”

line 142: “interface.Thus,” should be “interface. Thus,”

line 143: “are” should be “is”

line 147: “immunosensorsfabricated” should be “immunosensors fabricated”

line 148: “appliedfor” should be “applied for”

line 169: “at” should be “in”

line 179: “were” should be “was”

line 182: “amount” should be “number”

line 197: “multipex” should be “multiplex”

line 202: “tothe” should be “to the”

line 215: “biomarker” should be “biomarkers”

line 232: “cortisoldetection” should be “cortisol detection”

line 237: “alow” should be “a low”

line 237: “showedgood” should be “showed good ”

line 244: “thecortisol detectionin” should be “the cortisol detection in”

and so on…..

Response: Thank you for your careful review. We have carefully checked the manuscript and modified the language, which is shown in the revised manuscript.
(2) The review needs careful organization.

Response: We have carefully checked the manuscrpt and made some organizational modifications (mainly in the Introduction and Conclusion and outlook sections), which are shown in the revised manuscript.

Reviewer 3 Report

Recent Advances of MXene-based Electrochemical Immunosensors by Meiqing Yang et al.

 I went through the paper very carefully and thoroughly. Authors focused on the electrochemical immunosensors which is the largest class of affinity biosensing devices with strong practicability. In addition they used MXenes because in recent years, MXenes have become hotspot materials of electrochemical biosensors for their excellent properties, including large specific surface area, good electrical conductivity, high hydrophilicity and rich functional group. In this review, we firstly introduce the composition and structure of MXenes, as well as their properties relevant to the construction of biosensors.

There are a lot of shortcomings at this moment that has to be improved later for publication.

1-The paper contains interesting sciences in Immunosensors. The impact of the paper is going be good. Also, the quality of the research work presented in the paper is also good.

2-In general, ideas are well explained and understandable but, some tenses, linkers and grammar structures must be checked.

  1. Authors should obtain the novelty of this manuscript compared to published results?
  2. The authors should argue about the relevance of the temperature dependence by different methods of the coating because the temperature ha serious role in thermal studies.
  3. The Introduction does not provide sufficient background. The introduction does not explain the major contributions and novelty of this work. The significance of the proposed solution has not been summed up.
  4. There are many equations without references?

8- The constructive discussions are missing. As mentioned earlier, authors must make a comparative analysis with other similar solutions and back up their claims on how the proposed solution can be considered as high performing compared to others

9- How their results will be affected if they include energy loss in layers.

10- The novelty of this work should be stated explicitly in the text of the manuscript so that readers can get it easily.

11- Authors should compare their results with the published data and different results.

12- There a lot of published papers in this field, authors should be explained the new in these results in immunosensors.

13- Authors should be explained the distribution of electric fields with this structure as well as the equations related. It is will be very useful.

14- What is the role of MXene in your work?

15- Is the MXene affected by the light and if yes I hope authors calculate the refractive index of MXene in the related light region.

16- Why authors use MXene?

17- All figures, symbols, equations should be improved.

18- I am impressed to the manuscript title, really in fabrication, is it possible can MXene help in stability? Why?

19- Every term and structure in the proposed design (Cd0.5Zn0.5S/Ti3C2Tx MXene nanocomposite) should be clearly and correctly presented not to mislead the reader.

20- How this device can be stable with these kinds of materials especially with high temperature.

21- Is these measured results or simulated? What kind of simulation or fabrication method is performed?

22- This paper needs to make a brief explanation for the comparative analysis of the sensitivity selected at this concentration of MXene.

23-- In my opinion, due to the lack of experimental results, poor analysis of the properties of designed structure, low quality of the analysis of presented data, and lack of the sources for material properties used in this study, I suggest authors should revise this article carefully.

I wish to resend this paper after corrections and revise my comments

Author Response

(1) The paper contains interesting sciences in Immunosensors. The impact of the paper is going be good. Also, the quality of the research work presented in the paper is also good.

Response: Thanks for your comment on the paper.
(2) In general, ideas are well explained and understandable but, some tenses, linkers and grammar structures must be checked.

Response: Thank you for your suggestion, we have carefully checked the tense, connectors and grammatical structure of the paper and made some modifications. The modifications are shown in the revised manuscript.
(3) Authors should obtain the novelty of this manuscript compared to published results?

Response: We illustrate the novelty of this paper compared to the published related reviews, which is shown in the Introduction section of the revised manuscript.

(4) The authors should argue about the relevance of the temperature dependence by different methods of the coating because the temperature ha serious role in thermal studies.

Response: Thanks for your suggestion. However, this paper mainly focuses on the properties and roles of MXenes in electrochemical studies rather than their thermal studies. Besides, the discussion on the temperature dependence by different MXene coating methods has not been covered in previous MXene-based electrochemical immunosensors, so this topic is also not covered in this paper.

(5) The Introduction does not provide sufficient background. The introduction does not explain the major contributions and novelty of this work. The significance of the proposed solution has not been summed up.

Response: We have added some background to the Introduction section and explained the main contributions and novelty of this work by comparing with previous related work, and finally stated the purpose and significance of this work at the end of the Introduction section, the modifications are shown in the revised manuscript.

(6) There are many equations without references?

Response: Equations are not involved in this paper.

(8) The constructive discussions are missing. As mentioned earlier, authors must make a comparative analysis with other similar solutions and back up their claims on how the proposed solution can be considered as high performing compared to others

Response: We have added some constructive discussion to the Introduction section. In addition, the performance of sensors is discussed in Section 3 by introducing materials and sensor design.

(9) How their results will be affected if they include energy loss in layers.

Response: the work on MXene-based electrochemical immunosensors focuses on the electrochemical performance of MXenes and usually does not involve discussions of energy loss.

(10) The novelty of this work should be stated explicitly in the text of the manuscript so that readers can get it easily.

Response: Thanks again for your suggestion. We have added a description of the novelty of this work in the Introduction section.

(11) Authors should compare their results with the published data and different results.

Response: This is a review article rather than an experimental article, so there is no data comparison. In addition, we have added the differences from the related published work to the Introduction section, which are shown in the revised manuscript.

(12) There a lot of published papers in this field, authors should be explained the new in these results in immunosensors.

Response: There are a lot of published papers in the field of MXene-based sensors, but this paper focuses on the development of MXene-based electrochemical immunosensors, and is the first comprehensive and systematic review of MXene -based electrochemical immunosensors.

(13) Authors should be explained the distribution of electric fields with this structure as well as the equations related. It is will be very useful.

Response: Thanks for your suggestion. The topic of distribution of the electric field has not been mentioned in previous work on MXene-based electrochemical immunosensors, and thus the topic has also not been covered in this paper.

(14) What is the role of MXene in your work?

Response: MXene can be used as electrode-modified material, or detection signal molecule, or a carrier for detection signal molecule in the work.

(15) Is the MXene affected by the light and if yes I hope authors calculate the refractive index of MXene in the related light region.

Response: MXenes exhibit light absorption capacity, but this property is only considered in MXene-based optical nanoplatforms and photothermal immunoassays [1]. This article discusses MXene-based electrochemical immunosensors, focusing on the electrochemical properties of MXenes. Furthernore, the previous MXene-based electrochemical immunosensors also do not consider the effect of light, so the effect of light is also not covered in this paper.

(16) Why authors use MXene?

Response: Because MXene has become a hot material for the preparation of electrochemical biosensors due to its large specific surface area, high electrical conductivity, good hydrophilicity, abundant surface functional groups and other advantages in recent years.

(17) All figures, symbols, equations should be improved.

Response: The resolution of all the figures have been checked to ensure they meet the journal requirements and readability.

(18) I am impressed to the manuscript title, really in fabrication, is it possible can MXene help in stability? Why?

Response: MXenes are used in the fabrication of electrochemical immunosensors because they facilitate charge transfer, and facilitate loading biomolecules, etc., rather than help stabilize the sensor. In fact, MXenes are vulnerable in hot and humid environments. MXenes can usually maintain stability for several weeks or months [1, 2] under conventional storage conditions because their oxidation is a gradual process. Furthermore, the stability of MXenes can be successfully addressed by various strategies. First, studies have pointed out that water molecules and dissolved oxygen play crucial roles in the oxidation process, hence adding antioxidants such as L-ascorbic acid can effectively prolong the stability of MXenes. Second, surface modifications such as hydrophobic layer functionalized MXenes exhibit extraordinary sensing stability even in humid atmospheres. Finally, the hybridization of MXenes with other elements such as metal nanoparticles and enzymes also showed satisfactory stability compared to pristine MXenes [1].

(19) Every term and structure in the proposed design (Cd0.5Zn0.5S/Ti3C2Tx MXene nanocomposite) should be clearly and correctly presented not to mislead the reader.

Response: Thanks for your suggestion. Some term abbreviations for the above work are explained in Table1 at their first appearance, in addition, the explanation of d-Ti3C2Tx MXene is supplemented, which is shown in the revised manuscript.

(20) How this device can be stable with these kinds of materials especially with high temperature.

Response: It is reported that MXenes have high thermal stability [3], which can fully meet conventional testing requirements, so the temperature factor is usually not considered in the application of electrochemical immunosensors. Furthermore, if MXene-based materials need to be stable at very high temperature, MXenes need to be integrated with materials with excellent thermal insulation properties [4].

(21) Is these measured results or simulated? What kind of simulation or fabrication method is performed?

Response: The results mentioned in the text are all measured. The conventional preparation method of electrochemical immunosensor is performed.

(22) This paper needs to make a brief explanation for the comparative analysis of the sensitivity selected at this concentration of MXene.

Response: I am sorry for not getting your point, because the specific concentration of MXene is not mentioned in the manuscript.

(23) In my opinion, due to the lack of experimental results, poor analysis of the properties of designed structure, low quality of the analysis of presented data, and lack of the sources for material properties used in this study, I suggest authors should revise this article carefully.

Response: According to your suggestion, we have carefully revised the full manuscript.

References

[1] Li, X.; Lu, Y.; Liu, Q. Electrochemical and optical biosensors based on multifunctional MXene nanoplatforms: Progress and prospects. Talanta 2021, 235, 122726.

[2] Thenmozhi, R.; Maruthasalamoorthy, S.; Nirmala, R.; Navamathavan, R. Review—MXene Based Transducer for Biosensor Applications. J. Electrochem. Soc. 2021, 168, 117507.

[3] Kalambate, P.K.; Gadhari, N.S.; Li, X.; Rao, Z.; Navale, S.T.; Shen, Y.; Patil, V.R.; Huang, Y. Recent advances in MXene–based electrochemical sensors and biosensors. TrAC, Trends Anal. Chem. 2019, 120, 115643.

[4] Highly Compressible, Thermally Stable, Light-Weight, and Robust Aramid Nanofibers/Ti3AlC2 MXene Composite Aerogel for Sensitive Pressure Sensor. ACS Nano 2020, 14, 10633–10647.

Round 2

Reviewer 2 Report

Accept in present form

Reviewer 3 Report

All comments has been considered by authors